# Morpholine, Piperazine, and Piperidine Derivatives as Antidiabetic Agents

**DOI:** 10.3390/molecules29133043

**Published:** 2024-06-26

**Authors:** Darya Zolotareva, Alexey Zazybin, Anuar Dauletbakov, Yelizaveta Belyankova, Beatriz Giner Parache, Saniya Tursynbek, Tulegen Seilkhanov, Anel Kairullinova

**Affiliations:** 1School of Chemical Engineering, Kazakh-British Technical University, 59 Tole bi Str., Almaty 050000, Kazakhstan; zolotareva.2909@mail.ru (D.Z.); dayletbakovanuar@gmail.com (A.D.); belyankovae@gmail.com (Y.B.); erzhanovnasss@gmail.com (S.T.); anelkin.ka@gmail.com (A.K.); 2Facultad de Ciencias de la Salud, Universidad San Jorge, 50830 Zaragoza, Spain; bginer@usj.es; 3Laboratory of Engineering Profile NMR Spectroscopy, Sh. Ualikhanov Kokshetau University, 76 Abay Str., Kokshetau 020000, Kazakhstan; tseilkhanov@mail.ru

**Keywords:** antidiabetic drugs, diabetes, morpholine, piperazine, piperidine

## Abstract

Diabetes mellitus is a severe endocrine disease that affects more and more people every year. Modern medical chemistry sets itself the task of finding effective and safe drugs against diabetes. This review provides an overview of potential antidiabetic drugs based on three heterocyclic compounds, namely morpholine, piperazine, and piperidine. Studies have shown that compounds containing their moieties can be quite effective in vitro and in vivo for the treatment of diabetes and its consequences.

## 1. Introduction

Diabetes mellitus is an endocrine chronic disease characterized by high sugar levels in the blood. It occurs when the pancreas does not produce enough insulin or when the organism has insulin resistance [1]. The common symptoms of diabetes are thirst, polyuria, weight loss, blurred vision, hunger, numb or tingling hands or feet, fatigue, dry skin, sores that heal slowly, nausea, vomiting, or stomach pains. Type 1 diabetes symptoms can develop precipitously in a few weeks or months. It most often occurs at an early age. Type 2 diabetes develops slowly over several years and usually occurs in adults [1]. A total of 8.5% of the persons who were 18 years of age and older had diabetes in 2014. A total of 1.5 million deaths were directly related to diabetes in 2019, and 48% of these deaths occurred in those under the age of 70. Diabetes contributed to an additional 460,000 renal disease fatalities, and elevated blood glucose is responsible for almost 20% of the cardiovascular mortality [2]. Age-standardized diabetes mortality rates increased by 3% between 2000 and 2019. Diabetes-related death rates rose by 13% in lower-middle-income nations. In contrast, between 2000 and 2021, there was a 22% global decline in the likelihood of dying from any of the four major noncommunicable diseases (cancer, chronic respiratory diseases, diabetes, or cardiovascular diseases) between the ages of 30 and 70 (Figure 1) [2].

The major long-term consequences of diabetes relate to multiple vascular complications including retinopathy, nephropathy, neuropathy, atherosclerosis, angina pectoris, myocardial infarction, transient ischemic attack, strokes, peripheral arterial disease, and immune dysfunction [1].

Except for smooth muscle, insulin is the main hormone that controls the uptake of glucose from the blood into most body cells, including the liver, adipose tissue, and muscles, where insulin operates through IGF-1. Therefore, all types of diabetes mellitus are fundamentally caused by an inadequate supply of insulin or by the insensitivity of its receptors [3]. The breakdown of glycogen (glycogenolysis), the liver’s storage form of glucose; intestinal absorption of meals; and gluconeogenesis are the three main sources of glucose for the body. To control the body’s glucose levels, insulin is essential. Insulin can increase the transfer of glucose into fat and muscle cells as well as the process of gluconeogenesis. It can also stimulate the storage of glucose in the form of glycogen [4].

For type 1 diabetes, there is no proven preventive intervention [5]. Type 2 diabetes, which makes up 85–90% of the cases around the world, can frequently be avoided or delayed [6] by a balanced diet, staying physically active, and keeping a normal body weight [5]. More physical exercise (90 min or more per day) lowers the risk of diabetes by 28% [7]. Maintaining a diet high in whole grains and fiber, as well as picking healthy fats like the polyunsaturated fats found in fish, nuts, and vegetable oils are dietary adjustments proven to be useful in preventing diabetes. Diabetes can be prevented by limiting sugary drinks, consuming less red meat, and cutting back on other sources of saturated fat.

Some natural compounds contain chemical moiety which is beneficial in treating diabetes. These compounds are contained in aloe, mint, banaba, bitter melon, caper bush, cinnamon, cocoa, coffee, fenugreek garlic, guava, turmeric, tea, walnuts, shaggy bindweed, *Yerba mates*, *Bambusa tulda*, *Ficus bengalensis*, *Ferula orientalis*, *Gymnema sylvestre*, *Dioscorea japonica*, *Artemisia abyssinica*, *Phaseolus vulgaris*, *Datura quercifolia*, *Cassia fistula*, *Citrus aurantium*, *Ficus benghalensis*, *Polygonum aviculare*, *Allium tuncelianum*, *Astragalus brachycalyx*, *Ferulago stelleta*, and *Rhizophora mucronate* [8].

For instance, the thiazolidinedione–morpholine hybrid of vanillin (vanillin was isolated from a medicinal plant, *Polygonum aviculare*) (Figure 2) showed a good to moderate inhibition potential against α-glucosidase, α-amylase, and protein tyrosine phosphatase in vivo. However, it exhibited excellent in vitro inhibition of dipeptidyl peptidase-4 (DPP-4). This multitarget compound was synthesized by combining three pharmacophoric moieties into a single chemical entity that can modulate more than one target at the same time [9].

Piperine is an alkaloid found in various pepper varieties and the main one responsible for the pungent taste of black pepper (*Piper nigrum*) which also has a proven antidiabetic effect. Piperine contains piperidine moiety (Figure 3). The combination of piperine with a therapeutic dose of metformin (10 mg/kg + 250 mg/kg) in vivo showed a significantly more lowering of blood glucose level as compared to metformin alone on both the 14th and 28th day (*p* < 0.05) (in diabetic mice). Piperine in combination with a sub-therapeutic dose of metformin (10 mg/kg + 125 mg/kg) showed a significantly more lowering of blood glucose as compared to a control group and showed a greater lowering of blood glucose as compared to metformin (250 mg/kg) alone [10]. 

A study [11] found that after eight weeks of treatment with the aqueous extracts of black pepper (BP), turmeric (T), ajwa pulp (AP), ajwa seeds (AS), and their various combinations, the mean serum glucose, glycosylated hemoglobin, and insulin level of diabetic rats treated with the extracts significantly decreased when compared to the control group. By the conclusion of the trial, the mean blood glucose level in the BP + T + AP + AS (229.53 ± 14.33 mg/dL) and BP + T (233.86 ± 11.86 mg/dL) groups was considerably lower than in the other groups. The group that received treatment with BP + T + AP + AS had a considerably lower mean percentage of GHb overall. A non-significant variation in blood insulin levels was noted across the various treatment groups.

Berberine is an isoquinoline alkaloid, and is the major pharmacological component of the Chinese herb *Coptis chinensis* (traditional Chinese medicine). Recently, some research paid attention to the strong impact of this herb on glucose homeostasis and marked antidiabetic effects on both humans and rats. In [12], the authors presented piperidine-, piperazine-, and morpholine-substituted berberine (Figure 4). All the compounds in Figure 4 exhibit concentration-dependent insulin-resistant reversal actions that are superior to those of the positive control drug rosiglitazone. Compound 2 has the highest activity among them, with a 1.26-fold increase in sensitization compared to rosiglitazone. When the concentration is reduced to 1 mol/mL, the activity remains comparable to that of the positive control.

Despite the large number of natural substances, science is focused on finding synthetic compounds for the treatment of diabetes. Their number is already very large and continues to grow. These include metformin, gliquidone, nateglinide, phenformin, rosiglitazone, glimepiride, pioglitazone, glibenclamide, exenatide, mitiglinide, gliclazide, chlorpropamide, glipizide, acetohexamide, tolbutamide, dapagliflozin, dulaglutide, liraglutide, glyburide, canagliflozin, repaglinide, thiazolidinediones, biguanides, sulfonylureas, gliptins, deazaxanthine, pyrazole, pyrrolidine, oxindole, isatin, imidazole, benzimidazole, triazole, oxadiazole, thiazole, pyridine, piperazine, thiazolidinone, thiadiazole, benzofuran, benzoxazole, coumarin, flavone, piperidine, xanthone, and pyrimidine [8].

There are hundreds of synthetic compounds on the market. Medications for the treatment of type 1 diabetes include short-acting insulin (Humulin R U-100 (by Eli Lilly and Company, Indianapolis, IN, USA) and Novolin R FlexPen (Novo Nordisk, Bagsværd, Denmark)), rapid-acting insulin (inhaled insulin (Afrezza by MannKind Corporation, Westlake Village, CA, USA), insulin aspart (manufacturer Fiasp), insulin glulisine (Apidra by Sanofi, Paris, France), insulin lispro (Admelog by Sanofi), and insulin lispro-aabc (Lyumjev by Eli Lilly and Company)), intermediate-acting insulin (insulin isophane (Humulin N U-100 by Eli Lilly and Company)), long-acting insulin (insulin degludec (Tresiba by Novo Nordisk), insulin detemir (Levemir by Novo Nordisk), insulin glargine (Basaglar KwikPen by Eli Lilly and Company, Lantus by Sanofi), and insulin glargine-yfgn (Semglee-yfgn by Viatris Inc., Canonsburg, PA, USA)), and concentrated regular insulin (Humulin R U-500 by Eli Lilly and Company and Pramlintide (SymlinPen by AstraZeneca, Cambridge, UK)). [13]

Medications for treatment type 2 diabetes include the same insulin drugs as shown above and also α-glucosidase inhibitors (acarbose and miglitol (Glyset by Pfizer, New York, NY, USA)), Biguanides (metformin (Glumetza by Santarus Inc., San Diego, CA, USA and Riomet by Sun Pharmaceutical Industries Inc., Mumbai, India)), Dopamine-2 agonists (Bromocriptine (Cycloset by Veroscience and Parlodel by Sandoz Pharmaceuticals Corp., Basel, Switzerland)), Dipeptidyl peptidase-4 (DPP-4) inhibitors (alogliptin (Nesina by Takeda Pharmaceuticals, Tokyo, Japan) and alogliptin-metformin (Kazano by Takeda Pharmaceuticals)), Glucagon-like peptide-1 receptor agonists (GLP-1 receptor agonists) (dulaglutide (Trulicity), exenatide (Byetta by Amylin Pharmaceuticals, San Diego, CA, USA/Eli Lilly & Co.)), Sodium-glucose transporter (SGLT) 2 inhibitors (canagliflozin (Invokana by Janssen Pharmaceuticals, Beerse, Belgium)), Thiazolidinediones (rosiglitazone), and other (aspirin, medications for high cholesterol, and high blood pressure medications) [13].

Below we present an overview of three groups of these compounds, namely the compounds based on morpholine, piperazine, and piperidine.

Morpholine (Figure 5) is a well-known and widely used synton in medicinal chemistry. Its derivatives are used as analgesics, antioxidants, antiobesity medications, analeptics, anti-depressants, antibiotics, anticancer drugs, and anticoagulants [14]. 

Piperazine (Figure 5) is a compound having several applications in pharmaceuticals for humans; however, it is mostly used in veterinary medicine as antihelminthics [15].

Piperidine moiety (Figure 5) is a fundament in the production of drugs which are used as anticancer, antimicrobial, analgesic, anti-inflammatory, and antipsychotic agents [16].

## 2. Morpholine Derivatives as Antidiabetic Drugs

In [17], the authors reported about α-glucosidase inhibitors based on morpholine and piperazine structures (novel benzimidazole derivatives) which may be effective in type II diabetes (Figure 6). The α-Glucosidase effect is due to its ability to promote the rapid generation of blood glucose [14]. The most effective inhibitors inhibited the enzyme by 63% and 99%; moreover, these compounds showed antioxidant activity [17].

Khan et al. [18] also synthesized a series of 1-benzyl-3-((2-substitutedphenyl)amino)-2-oxoethyl)-2-(morpholinomethyl)-1H-benzimidazol-3-ium chloride (Figure 7) and checked them for α-glucosidase inhibitory activity in vitro. The compound 1-benzyl-3-(2-((4-bromophenyl)amino)-2-oxoethyl)-2-(morpholinomethyl)-1H-benzo[d]imidazol-3-ium chloride possessed potent inhibitory potential against the α-glucosidase enzyme. Furthermore, compounds 1-benzyl-3-(2-((3-nitrophenyl)amino)-2-oxoethyl)-2-(morpholinomethyl)-1H-benzo[d]imidazol-3-ium chloride, 1-benzyl-3-(2-((2-methylphenyl)amino)-2-oxoethyl)-2-(morpholinomethyl)-1H-benzo[d]imidazol-3-ium chloride, 1-benzyl-3-(2-((4-nitrophenyl)amino)-2-oxoethyl)-2-(morpholinomethyl)-1H-benzo[d]imidazol-3-ium chloride, 1-benzyl-3-(2-((2-methoxyphenyl)amino)-2-oxoethyl)-2-(morpholinomethyl)-1H-benzo[d]imidazol-3-ium chloride, and 1-benzyl-3-(2-((2-nitrophenyl)amino)-2-oxoethyl)-2-(morpholinomethyl)-1H-benzo[d]imidazol-3-ium chloride also showed moderate to good α-glucosidase inhibitory activity.

The series of 4-(5-fluoro-2-substituted-1H-benzimidazol-6-yl)morpholine derivatives showed remarkable α-glucosidase inhibitory potentials [19]. A compound shown in Figure 8, which has a methoxy group on the phenyl ring, was found to be the most active molecule of the series. It was observed from in vitro and in silico studies that electron-donating groups such as methyl and methoxy on phenyl ring and increased conjugative effect played a significant role in the inhibition, which could be a promising lead for future investigation.

These heterocycles do not always act as active (pharmacophore) centers but may be responsible for the orientation of the molecule in the enzyme cavity. In [19], docking simulation of yeast isomaltase showed that hydrogen bond interactions occurred between oxygen atoms on the morpholine moiety of ligand and the backbone of arginine (ARG315). Docking simulation of the human lysosomal acid α-glucosidase crystal structure of the compound in Figure 8 (which was the most active compound in the series) also showed that this compound formed a hydrogen bond with the ARG275 residue and the oxygen atom of morpholine moiety and was connected to a receptor site. A study of α-glucosidase inhibitory activity [20] displays that the addition of morpholine moiety has a positive effect on inhibition properties in comparison with the structures without morpholine.

The optically active derivatives of 1,4-morpholin-2,5-dione (Figure 9) [21] also showed inhibitory activities against α-glucosidases (from both baker’s yeast and *Bacillus stearothermophilus*). The extension of the side chain by the introduction of a CH_2_OBn group increases the biological activity. The compounds (3*R*,6*S*,20*S*,30*R*)-3-(40-Benzyloxy-20,30-dihydroxybutyl)-6-methyl-4-[(S)-phenethyl]-morpholine-2,5-dione, (3*R*,6*S*,20*R*,30*S*)-3-(40-Benzyloxy-20,30-dihydroxybutyl)-6-methyl-4-[(*S*)-phenethyl]-morpholine-2,5-dione, and (3*S*,6*R*,20*S*,30*R*)-3-(40-Benzyloxy-20,30-dihydroxybutyl)-6-methyl-4-[(S)-phenethyl]-morpholine-2,5-dione are the most active inhibitors of the series towards α-glucosidases (both from baker’s yeast and with *Bacillus stearothermophilus*).

Recent in vitro research [22] reported that piperazinyl- and morpholinyl-quinaline derivatives (Figure 10) had a high ability to inhibit some metabolic enzymes, such as hCA I and II (58.26 ± 9.36–144.37 ± 19.03 µM), AChE (201.16 ± 30.84 and 198.27 ± 37.31 µM), BChE (245.73 ± 51.28 and 204.65 ± 19.26 µM), and α-glycosidase (584.20 ± 62.51–1023.16 ± 103.27 µM), compared with the standard compounds acetazolamide, tacrine, and acarbose and can be used for the treatment of some diseases such as gastric and duodenal ulcers, glaucoma, mountain sickness, epilepsy, osteoporosis and neurological disorders, alternative Alzheimer’s disease, and type-2 diabetes mellitus.

Morpholine-substituted thiadiazoles (Figure 11) showed histamine H3 receptor antagonists (in vivo), which means it can be used as obesity and type 2 diabetes treatment [23]. The 4-(5-([1,4′-bipiperidin]-1′-yl)-1,3,4-thiadiazol-2-yl)-2-(pyridin-2-yl)morpholine leads to reducing the non-fasting glucose levels, and also dose-dependently blocks the increase in HbA1c.

Morpholino thiazolyl-2,4-thiazolidinediones (2,4-TZD N-acetic acid, acetic acid, ethyl ester, benzyl, and phenacyl derivatives containing morpholinothiazole ring) (Figure 12) were tested in vitro comparing with glibenclamide for their insulinotropic activities in INS-1 cells and had positive effects with respect to insulin release and increased glucose uptake (they possess pancreatic and extrapancreatic effects) [24].

In 2022, the authors of [25] discovered remarkable antidiabetic in vivo activity of (4-{[1-({3-[4-(trifluoromethyl)phenyl]-1,2,4-oxadiazol-5-yl}methyl)piperidin-3-yl]methyl}morpholine (Figure 13), which acts as positive allosteric modulator (PAM) of GLP-1R (glucagon-like peptide-1 receptor), which helps to reduce food intake in obesity and to improve glucose handling in normal and diabetic patients. Moreover, there were no observed off-targeted activities.

The hybrid structure of sulfonamide-1,3,5-triazine-thiazole containing morpholine structure possesses (Figure 14) dipeptidyl peptidase 4 (DPP-4) selective inhibition activity (in vivo), which offers blood glucose-lowering effect [26]. It also showed an improvement of blood glucose level in a dose-dependent manner via significant improvement of insulin level and antioxidant enzyme systems. It has been clearly depicted that compounds containing electron-withdrawing groups showed more pronounced inhibitory activity in comparison to the electron-donating substituent. Moreover, it has been also found that the presence of additional aromaticity does not influence the activity.

Another study [27] of 1,3,5-triazines containing morpholine fragments (Figure 15) presented novel DPP-4 inhibitors which are devoid of any cardiac toxicity. The most active compound showed in vivo DPP-4 inhibition accompanied by a blood glucose lowering effect in the experimental subject and showed improvement in the oxidative system in rats. 

## 3. Piperazine Derivatives as Antidiabetic Agents

2-Furoic piperazide derivatives were synthesized [28] and were tested as α-glucosidase, acteylcholinesterase, and butyrylcholinesterase inhibitors through molecular docking study while their cytotoxicity was profiled through hemolytic activity. {4-[(3,5-dichloro-2-hydroxyphenyl)sulfonyl]-1-piperazinyl}(2-furyl)methanones molecules (the most promising molecule showed in Figure 16) exhibited very promising enzyme inhibitory potentials against α-glucosidase, AChE, and BChE enzymes. Additionally, a little cytotoxicity was noted for the compounds that were synthesized. Thus, it was determined from the present study that a few of the compounds are worthy of being considered as safe and effective therapeutic agents for the treatment of Alzheimer’s disease and type 2 diabetes.

A study [29] was conducted on 4-(dimethylaminoalkyl)piperazine-1-carbodithioate derivatives (the most promising compound is shown in Figure 17) with various substitutions on their benzene ring as α-glucosidase inhibitors. Enzyme kinetics, molecular dynamics simulations, and docking studies have been carried out and showed promising therapeutic prospects due to their lipophilic nature in addition to their α-glucosidase inhibitory activity.

β-Carboline derivatives containing piperazine moieties (Figure 18) were synthesized and evaluated for their α-glucosidase inhibitory activity [30]. Especially, one of the compounds presented obvious α-glucosidase inhibitory activity (IC_50_: 8.9 ± 0.2 μM), ~69 folds stronger than acarbose (IC_50_: 610.7 ± 0.1 μM). Cell cytotoxicity assay ascertained low cytotoxicity.

β-carboline derivatives are known for their ability to inhibit α-glucosidase activity [30,31]. These compounds typically contain a planar aromatic structure that can interact with the enzyme’s active site through π-π interactions. The presence of a piperazine moiety in β-carboline derivatives further enhances their inhibitory activity by providing additional interactions with the enzyme. The piperazine moiety may form hydrogen bonds or electrostatic interactions with specific residues within the enzyme’s active site, thereby stabilizing the enzyme–inhibitor complex.

Novel 6-(4-substitue-piperazin-1-yl)-2-aryl-1H-benzimidazole derivatives starting from 5-(4-substitue-piperazin-1-yl)-2-nitroaniline with different aldehydes (example compound is showed in Figure 19) showed within in vitro evaluations α-amylase and α-glucosidase inhibitory activities, as well as their antioxidant properties (DPPH radical scavenging capabilities) [32]. The results demonstrated that all the synthesized analogs exhibited the significant inhibition of both α-glucosidase and α-amylase potential between IC_50_ = 0.85 ± 0.25–29.72 ± 0.17 µM and IC_50_ = 4.75 ± 0.24–40.24 ± 0.10 µM, respectively, in comparison to the standard acarbose (IC_50_ = 14.70 ± 0.11 μM).

In 1999 [33], a series of 1-benzyl-4-alkyl-2-(4′,5′-dihydro-1′H-imidazol-2′-yl)piperazines were explored as antidiabetic agents (in vitro and in vivo). The most active compounds were 1,4-diisopropyl-2-(4′,5′-dihydro-1′H-imidazol-2′-yl)piperazine (Figure 20a), and 1,4-diisobutyl-2-(4′,5′-dihydro-1′H-imidazol-2′-yl)piperazine (Figure 20b) which greatly improved glucose tolerance without side effects or hypoglycemic effects. During this research, it was found that the presence of an unsubstituted imidazoline ring is a necessary pharmacophore for potential antihyperglycemic properties.

The investigation [34] of aryl piperazines and especially the most effective compound (Figure 21) claimed that these compounds promote glucose uptake and inhibit NADH:ubiquinone oxidoreductase, easily metabolize, avoid rapid excretion, have a short half-life, and have good tissue distribution (in vitro DMPK and in vivo PK studies). These compounds can be potential candidates for type 2 diabetes mellitus treatment. 

According to the study [34], a piperazine ring is best for activity, but substituting a piperidine ring does not significantly lessen activity. According to the findings, aryl piperazine is not enough to promote glucose absorption; instead, a more complex molecule is needed. Biological activity requires the presence of a p-CF_3_ aryl piperazine pattern, which has been shown to be more resistant to modifications in the thiophene heterocycle and alkyl chain. Overall, the study points to the potential efficacy of aryl piperazines, such as the compound shown in Figure 21, as diabetes therapy agents.

Piperazine-derived compounds with aryl piperazine moieties (Figure 22) [35] have been tested for DPP-4 inhibitory activity in vitro and antihyperglycemic, antidyslipidemic, and insulin reversal activities in vivo. The results showed that the compound in Figure 22 exhibited better in vitro inhibitory activity than the reference inhibitor and moderate in vivo antihyperglycemic, antidyslipidemic, and insulin resistance reversal activities as compared to the standard antidiabetic drug Sitagliptin. Molecular docking studies have also exhibited the good binding affinity of the compound at the active site of DPP-4 complementing the biological activity. This compound exhibited high mouse and human serum protein binding, and it had moderate bioavailability (21.4%). Therefore, the piperazine-derived constrained compound in Figure 22 has remarkable promise for further exploration as a potential small-molecule DPP-4 inhibitor.

Piperazine sulphonamide derivatives (the most active compound is demonstrated in Figure 23) have shown activity against DPP-4 enzyme in vitro and decreased serum glucose levels during in vivo tests [36].

In vitro biological evaluation study showed DPP-4 inhibitory activity for the sulfonyl piperazine derivatives (Figure 24) ranging from 19% to 30% at 100 μM concentration [37]. Furthermore, the in vivo hypoglycemic activity of 1,4-bis(4-fluorophenylsulfonyl)piperazine was evaluated using streptozotocin-induced diabetic mice. It was found that 1,4-bis(4-fluorophenylsulfonyl)piperazine significantly decreased the blood glucose level when the diabetic group treated with 1,4-bis(4-fluorophenylsulfonyl)piperazine was compared to the control diabetic group. Quantum-Polarized Ligand Docking (QPLD) studies demonstrate that these compounds fit the binding site of the DPP-4 enzyme and form H-bonding with the backbones of R125, E205, E206, K554, W629, Y631, Y662, R669, and Y752. 

A meta-analysis [38] presented data from several research studies of irbesartan combined Piperazine ferulate (PF) in the treatment of diabetic nephropathy (DN) (Figure 25). It was found that PF effectively improved the regular treatment of DN with irbesartan. The difference in adverse effects between the two groups (with and without PF) was not statistically significant, which means that adding PF to the main treatment does not pose any risks to the health of patients. In this study, the combination could control the levels of blood glucose by reducing the levels of FPG, 2 h PG, and HbA1c. PF combined with irbesartan plays a protective role in renal function by decreasing Scr, 24 h urinary protein, UAER, b2-MG, and BUN content. 

## 4. Piperidine Derivatives as Antidiabetic Agents

The well-known antidiabetic drug of the DPP-4 class alogliptin (Figure 26) contains piperidine moiety [39]. The in vivo study demonstrated great (IC50 < 10 nM) inhibition of DPP-4. It has over 10,000 times selectivity over the similarly related serine proteases DPP-8 and DPP-9. Furthermore, there was a strong association seen between the compound’s plasma concentration and the degree of DPP-4 inhibition in rats. The compound also illustrated enhanced plasma insulin levels and improved glucose tolerance in female Wistar fatty rats in a dose-dependent manner. Alogliptin has extremely positive outcomes from a safety pharmacology screen profile. GLP toxicity experiments in rats and dogs showed that the drug was well tolerated. Alogliptin showed human PK-PD appropriate for once-daily dosage in phase I human studies [39]. A multicentre, randomized, double-blind, placebo-controlled, phase 3 study of aliogliptin in patients with type 2 diabetes mellitus has been carried out [40]. In comparison to placebo, alogliptin as monotherapy resulted in a substantially higher reduction in HbA1c from baseline to week 16 (−0.58%; 95% confidence interval [CI] −0.78%, −0.37%; *p* < 0.001). Alogliptin used in conjunction with metformin or pioglitazone significantly reduced HbA1c levels when compared to placebo, with values of −0.69% (95% CI −0.87%, −0.51%; *p* < 0.001) and −0.52% (95% CI −0.75%, −0.28%; *p* < 0.001), respectively. Compared to a placebo, alogliptin significantly reduced fasting plasma glucose (FPG; *p* ≤ 0.004) and increased the proportion of patients who reached the ≤6.5% and ≤7.0% HbA1c goal (*p* ≤ 0.003). Not a single treatment group showed signs of weight increase. Both the alogliptin and placebo groups had comparable rates of patients experiencing drug-related and treatment-emergent side effects. Two patients in the placebo group and four individuals receiving alogliptin had mild to moderate hypoglycemia. Used as monotherapy, as an adjuvant to metformin, or as an adjuvant to pioglitazone, alogliptin 25 mg once daily decreased HbA1c and FPG and improved clinical response compared to the placebo. Alogliptin therapy was favorably received. 

The study [41] of the sulfonamide derivatives of piperidine (Figure 27) showed that the presence of nitrile group gives to compounds DPP-4 inhibition. The sulfonamide derivatives of piperidine-3-carbonitrile and pyrrolidine-2-carbonitrile showed great activity comparable with the standard drug Vildagliptin via in silico and in vitro studies. 

Among other things, in [41] it was reported that better inhibition was demonstrated by the small molecules of the sulfonamide derivatives of piperidine-3-carbonitrile and pyrrolidine-2-carbonitrile, which is equivalent to the standard (Vildagliptin). Additionally, DPP-IV inhibition was shown by the sulfonamide derivatives of piperidine-3-carbonitrile and pyrrolidine-2-carbonitrile to the same value as Vildagliptin. This study’s intriguing finding is that the derivative of piperidine-3 carboxylic acid exhibits five times the potency of the derivative from L-proline.

Piperidine-substituted chalcones [42] displayed α-amylase inhibitory and radical scavenging (DPPH and ABTS) activities (in vitro). Among all the studied compounds, the compound in Figure 28a showed the highest α-amylase inhibitory activity IC_50_ = 9.86 ± 0.03 μM compared with acarbose IC_50_ = 13.98 ± 0.11 μM. Among the compounds, the derivative in Figure 28b displayed the best radical scavenging activity IC_50_ = 25.40 ± 0.17 μM (DPPH) and 27.61 ± 0.16 μM (ABTS) comparable to that of standard ascorbic acid. Among other things, it must be mentioned that chalcones (1,3-diphenyl-2-propen-1-ones) are flavonoid-class natural compounds, which in themselves are of great interest for medicinal chemistry [43].

One more chalcone derivative of piperidine has shown the increased absorption of glucose along with the α-amylase, α-glucosidase, and sucrase enzyme inhibition [44]. All the synthesized compounds demonstrated high α-amylase inhibition (in vitro) at 1000 μg/mL (from 96.62% to 85.82%) compared with acarbose (83.28% at a concentration of 1250 μg/mL). The in vitro α-glucosidase inhibitory activity of the tested chalcone derivatives of piperidine also displayed promising results. The α-glucosidase inhibitory activity of all the compounds was higher than standard acarbose (69.54%); the highest percentage was gained by compound a (Figure 29a) and compound b (Figure 29b) (95.78% and 94.46%, respectively). The in vitro antioxidant activity of different compounds was estimated using DPPH, ABTS, Nitric oxide, SOD, hydrogen peroxide, and LPO. The synthesized compound b (86.37%), compound c (83.57%), and compound d (80.87%) (Figure 29) demonstrated higher activity than standard drugs (64.71%); the other substances showed moderate activity. In comparison to chalcones, the piperidine nucleus has a greater influence on biological activities due to its involvement in inhibiting the action of test compounds [44].

A novel piperidine derivative 4-methyl-1-({2,3,5,6-tetramethyl-4-[(4-methylpiperidinyl)methyl]phenyl}methyl)piperidine (Figure 30) showed antidiabetic (against α-amylase), antioxidant, and DNA-binding activities [45]. This compound showed an inhibitory activity of 97.30% and acarbose showed an inhibition of 61.66% against the α-amylase enzyme in vitro. In the DPPH radical test in vitro, the compound displayed moderate activity compared with the ascorbic acid. The in silico ADMET assessment results reveal that the piperidine derivative has a lower toxicity potential than the standard compound.

Piperidine-derived sulfonamides were evaluated for chemical reactivity, donor–acceptor interactions, the density of states, electrostatic potential, and molecular docking (in silico). Compounds a, b, and c (Figure 31) showed that these compounds fitted tightly to the amino acid residue-binding sites of insulin-inhibiting protein receptors (7M17) [46].

Piperidine amide and urea derivatives (an example compound is shown in Figure 32) were tested for potential 11b-HSD1 (11b-hydroxysteroid dehydrogenase isozymes) inhibitory activity. They showed good in vivo activity and reduced fasting and non-fasting blood glucose. On the other side, these compounds had a high lipophilic nature which means poor metabolic stability. Another major concern is the extreme selectivity against the human 11β-HSD2 enzyme [47].

## 5. Conclusions

The global trend towards an increase in the incidence of diabetes mellitus, as well as a decrease in the age of the diseased, dictates the need to look for new synthetic antidiabetic drugs with greater efficacy and minimal side effects. Our review article proves that morpholine, piperazine, and piperidine derivatives are promising potential antidiabetic agents. The conjugation of morpholine, piperazine, and piperidine moieties with natural molecules (for example berberine or vanillin or piperine) showed high potential to represent a large field for research. Another way of the search for new antidiabetic compounds is a combination of 2 or 3 pharmacological moieties among morpholine, piperazine, and piperidine in one molecule or the introduction of them into the compounds with bioactivities like antioxidant or anticancer for the simultaneous treatments of diabetes and the related diseases. For example, the benzimidazole derivatives of piperazine and morpholine and β-carboline derivatives containing a piperazine moiety exhibit α-glucosidase inhibitory activity due to their structural features that facilitate interactions with the enzyme’s active site. These interactions can include π-π interactions, hydrogen bonding, and electrostatic interactions, collectively contributing to the inhibition of α-glucosidase activity. This review article demonstrates the huge potential of the creation of antidiabetic or multifunctional drugs based on morpholine, piperazine, and piperidine moieties.

## Figures and Tables

**Figure 1 molecules-29-03043-f001:**
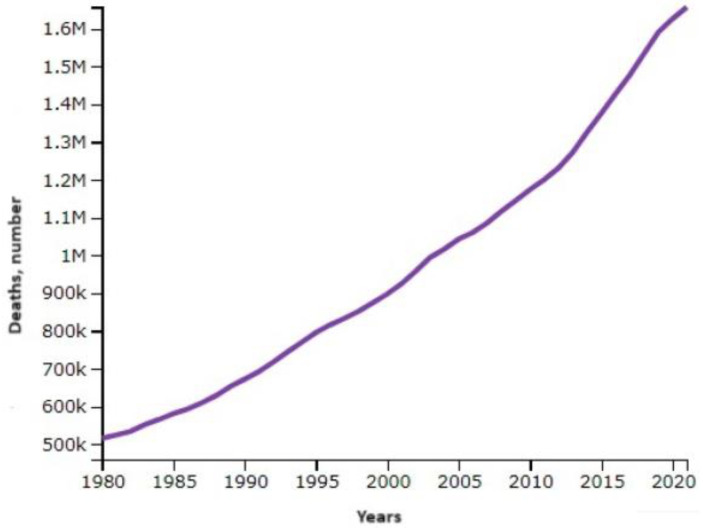
Deaths from diabetes mellitus from 1990 to 2021.

**Figure 2 molecules-29-03043-f002:**
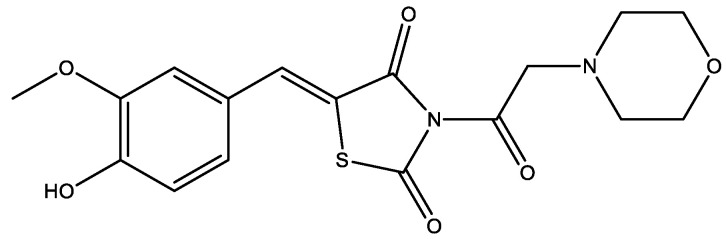
The thiazolidinedione–morpholine hybrid of vanillin.

**Figure 3 molecules-29-03043-f003:**
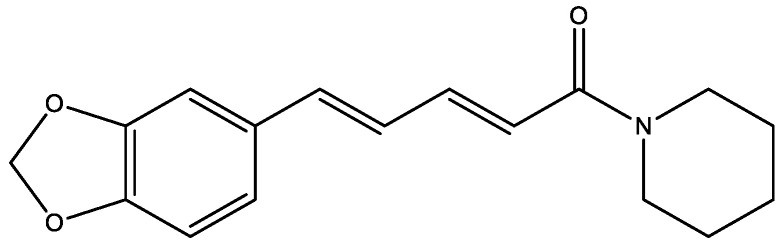
Piperine molecule.

**Figure 4 molecules-29-03043-f004:**
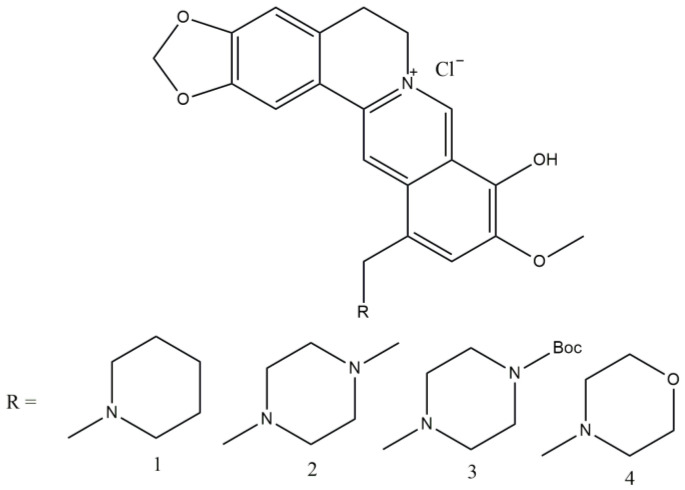
Piperidine-, piperazine-, and morpholine-substituted berberine.

**Figure 5 molecules-29-03043-f005:**
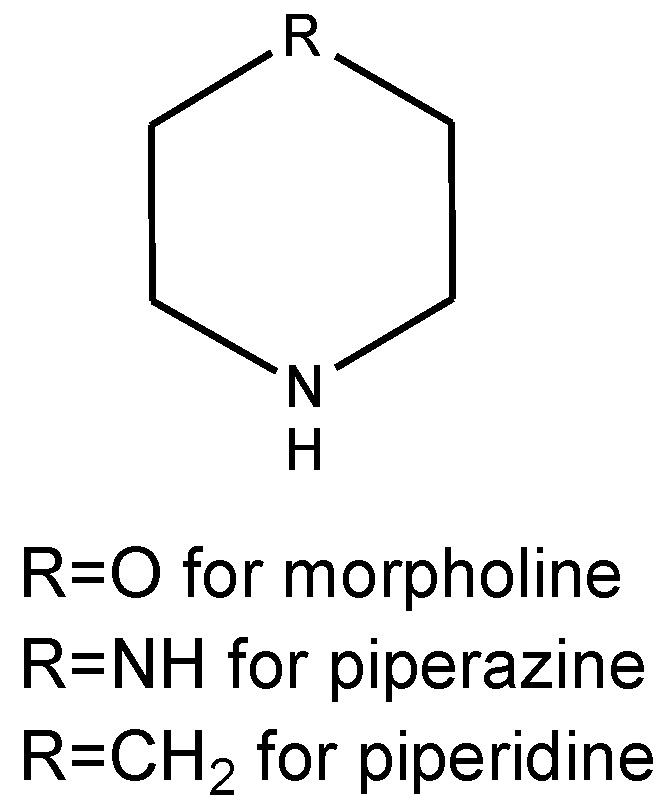
Structure of morpholine, piperazine, and piperidine moieties.

**Figure 6 molecules-29-03043-f006:**
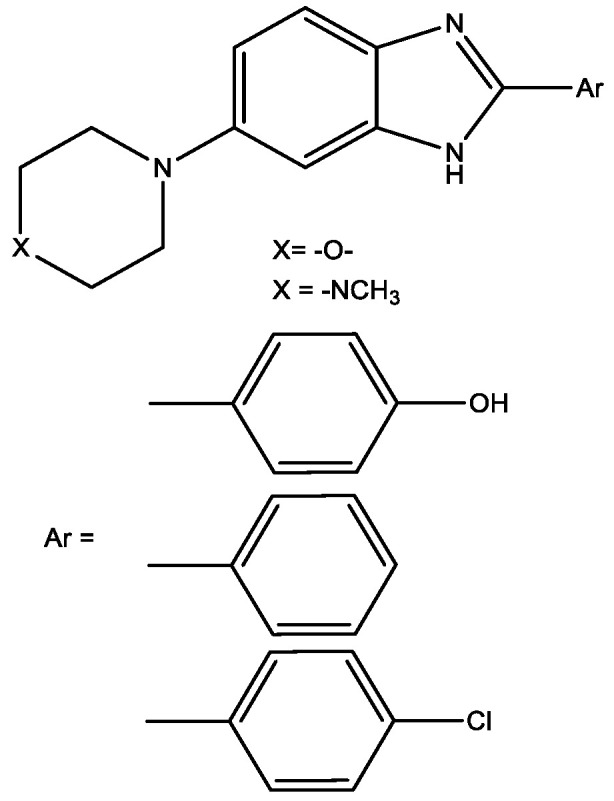
Benzimidazole derivatives as α-glucosidase inhibitors.

**Figure 7 molecules-29-03043-f007:**
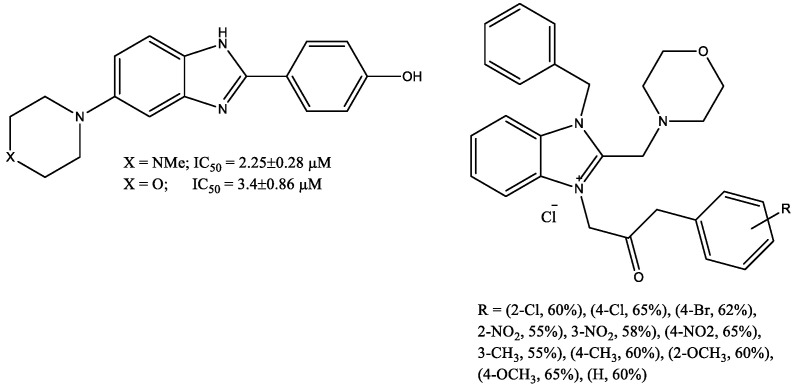
N-methylmorpholine-substituted benzimidazolium salts with α-glucosidase inhibitory activity.

**Figure 8 molecules-29-03043-f008:**
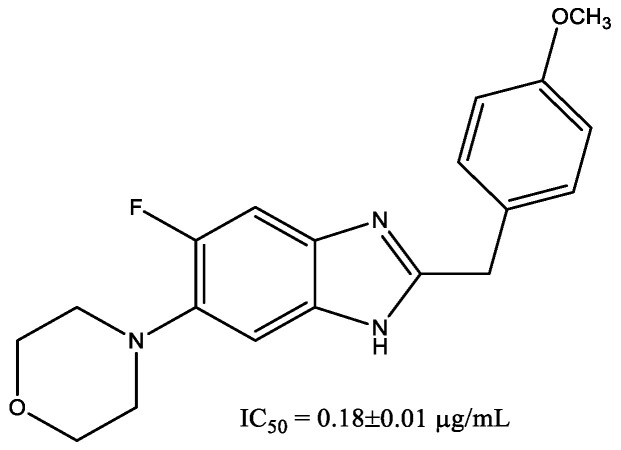
4-(5-Fluoro-2-(4-methoxybenzyl)-1H-benzo[d]imidazol-6-yl)morpholine as most effective compound of series.

**Figure 9 molecules-29-03043-f009:**
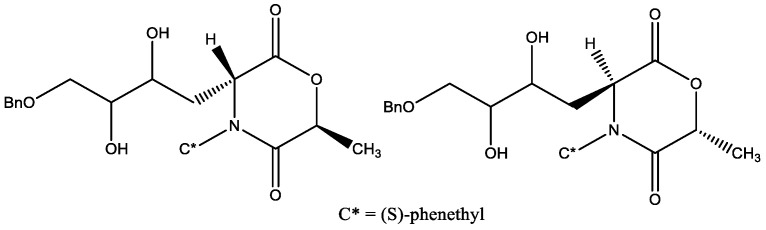
Optically active 1,4-morpholin-2,5-dione derivatives.

**Figure 10 molecules-29-03043-f010:**
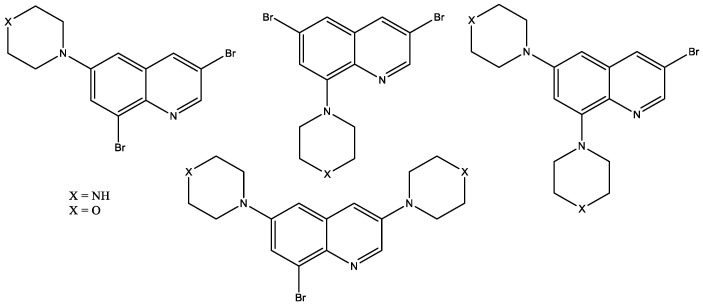
Piperazinyl- and morpholinyl-quinaline derivatives.

**Figure 11 molecules-29-03043-f011:**
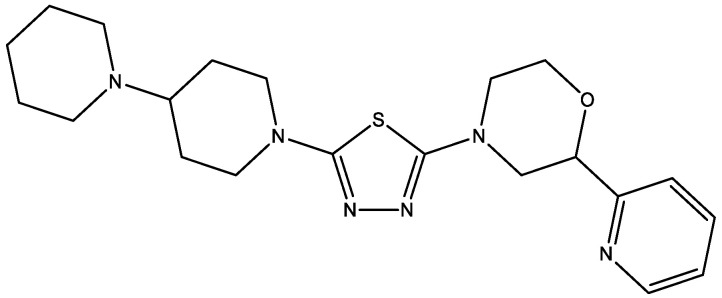
Example of morpholine-substituted thiadiazole derivative.

**Figure 12 molecules-29-03043-f012:**
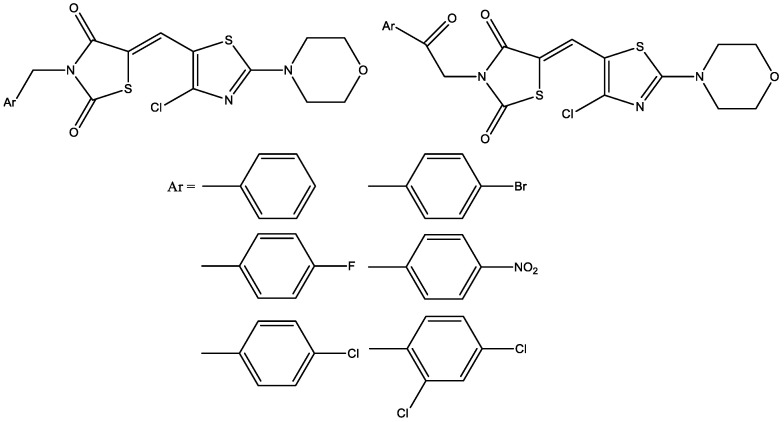
Morpholino thiazolyl-2,4-thiazolidinediones.

**Figure 13 molecules-29-03043-f013:**
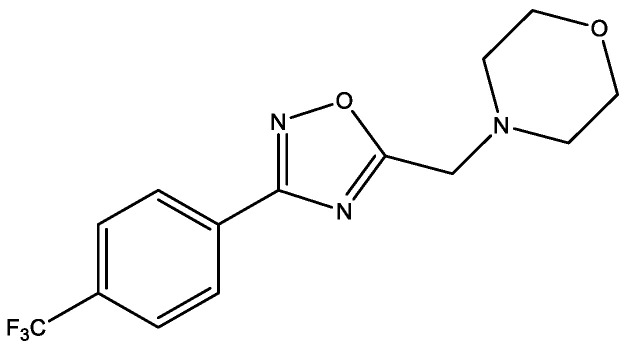
(4-{[1-({3-[4-(Trifluoromethyl)phenyl]-1,2,4-oxadiazol-5-yl}methyl)piperidin-3-yl]methyl}morpholine.

**Figure 14 molecules-29-03043-f014:**
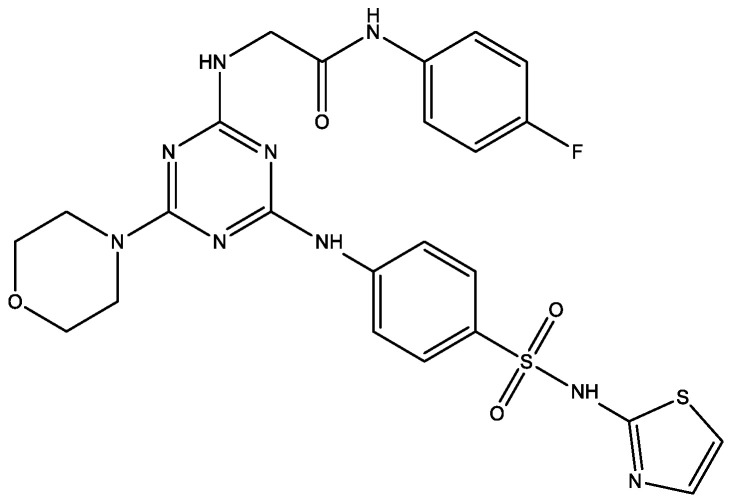
Sulfonamide-1,3,5-triazine-thiazole containing morpholine moiety.

**Figure 15 molecules-29-03043-f015:**
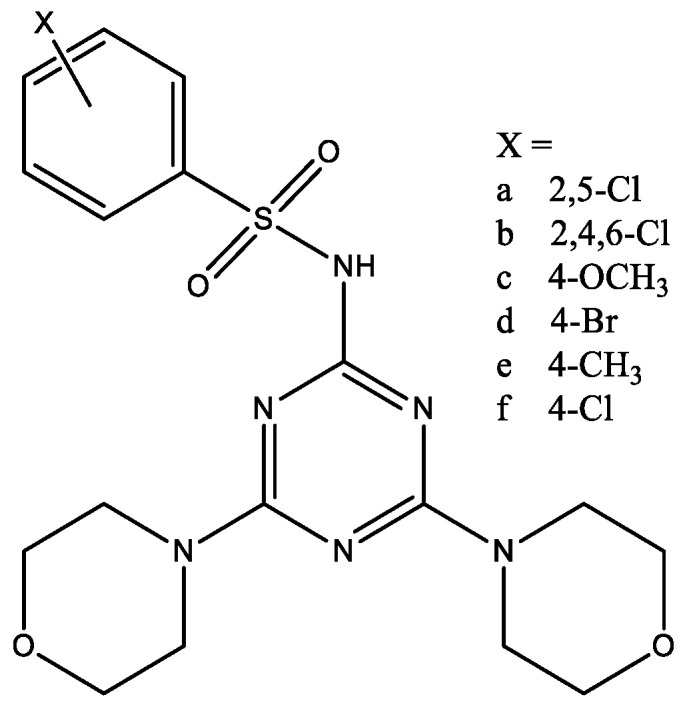
1,3,5-triazines containing two morpholine fragments.

**Figure 16 molecules-29-03043-f016:**
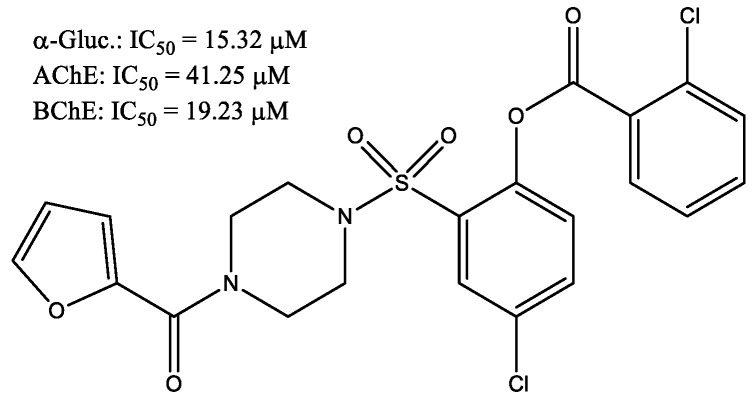
2,4-Dichloro-6-{[4-(2-furoyl)-1-piperazinyl]sulfonyl}phenyl 2-chlorobenzoate as potential α-glucosidase, AChE, and BChE enzyme inhibitor.

**Figure 17 molecules-29-03043-f017:**
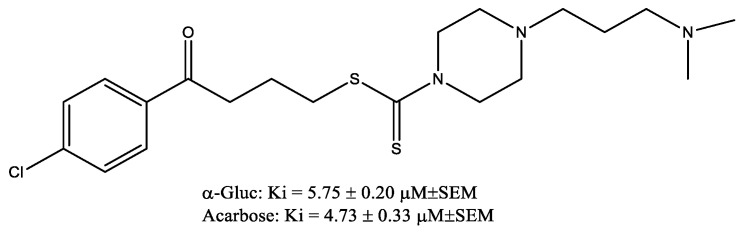
4-(dimethylaminoalkyl)piperazine-1-carbodithioate derivative.

**Figure 18 molecules-29-03043-f018:**
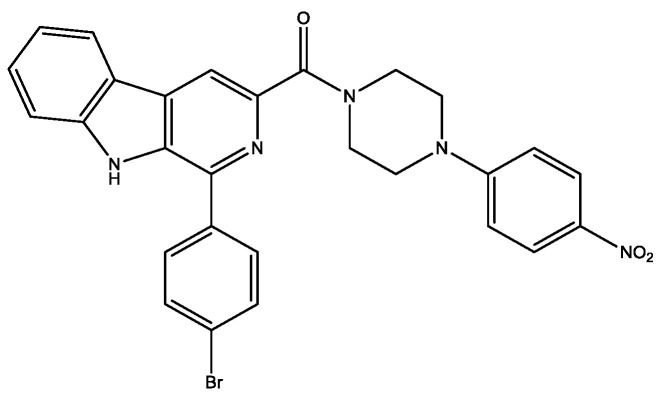
β-carboline derivative containing piperazine moiety with α-glucosidase inhibitory activity.

**Figure 19 molecules-29-03043-f019:**
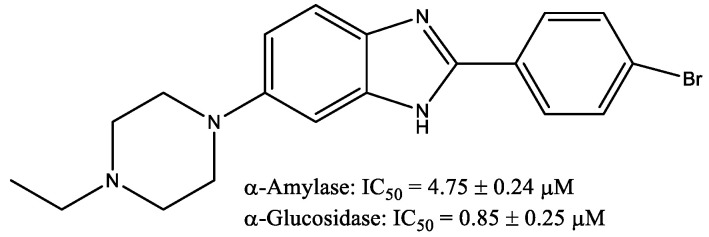
6-(4-Substitue-piperazin-1-yl)-2-aryl-1H-benzimidazole derivative.

**Figure 20 molecules-29-03043-f020:**
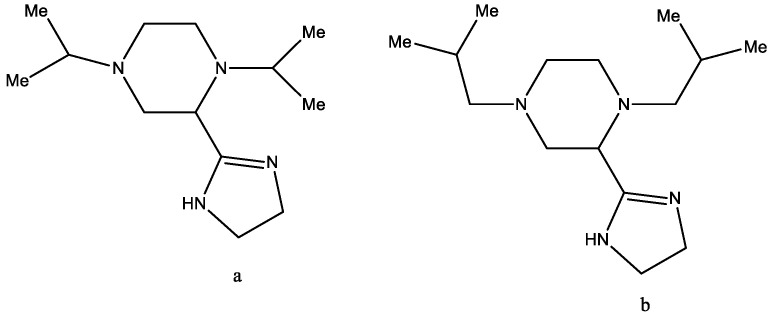
1,4-Diisopropyl-2-(4′,5′-dihydro-1′H-imidazol-2′-yl)piperazine (**a**) and 1,4-diisobutyl-2-(4′,5′-dihydro-1′H-imidazol-2′-yl)piperazine (**b**).

**Figure 21 molecules-29-03043-f021:**
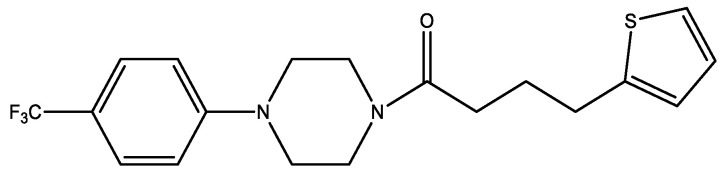
Structure of piperazine-based compound promoting glucose uptake.

**Figure 22 molecules-29-03043-f022:**
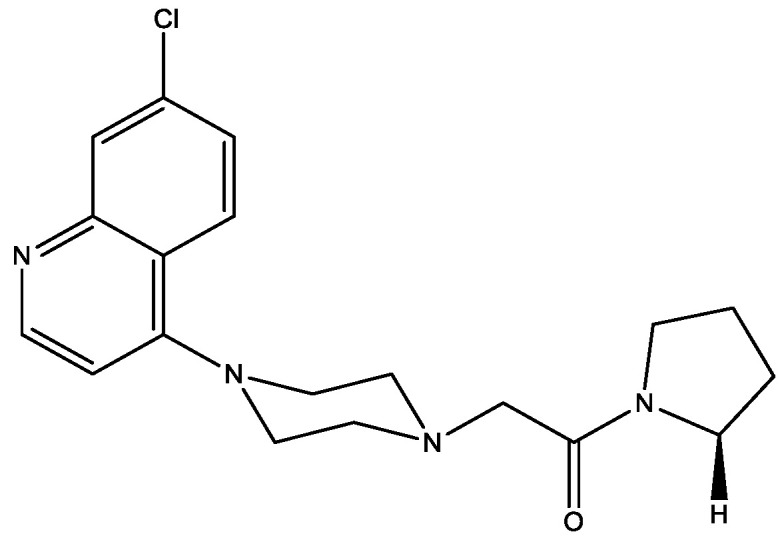
Piperazine-derived constrained compound as DPP-4 inhibitor.

**Figure 23 molecules-29-03043-f023:**
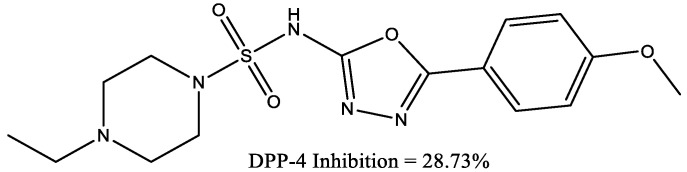
4-Ethyl-N-(5-(4-methoxyphenyl)-1,3,4-oxadiazol-2-yl) piperazine-1-sulfonamide.

**Figure 24 molecules-29-03043-f024:**
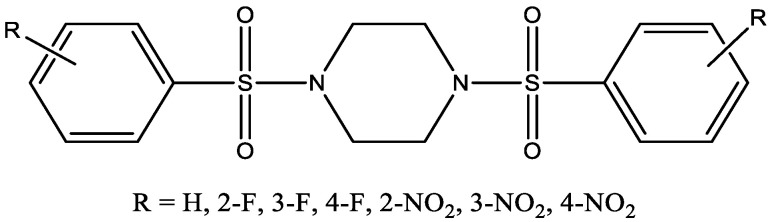
Sulfonyl piperazine derivatives as DPP-4 inhibitors.

**Figure 25 molecules-29-03043-f025:**
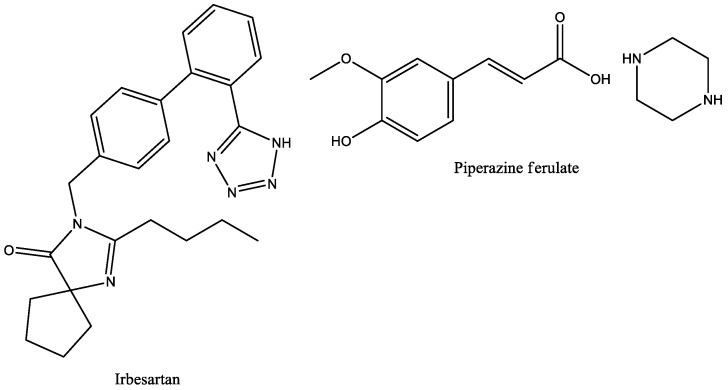
Structural formulae of irbesartan and piperazine ferulate (PF).

**Figure 26 molecules-29-03043-f026:**
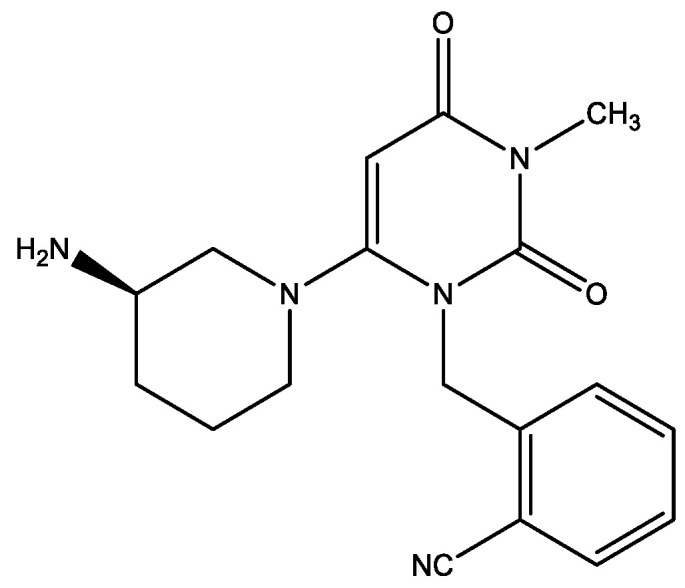
Structural formula of alogliptin.

**Figure 27 molecules-29-03043-f027:**
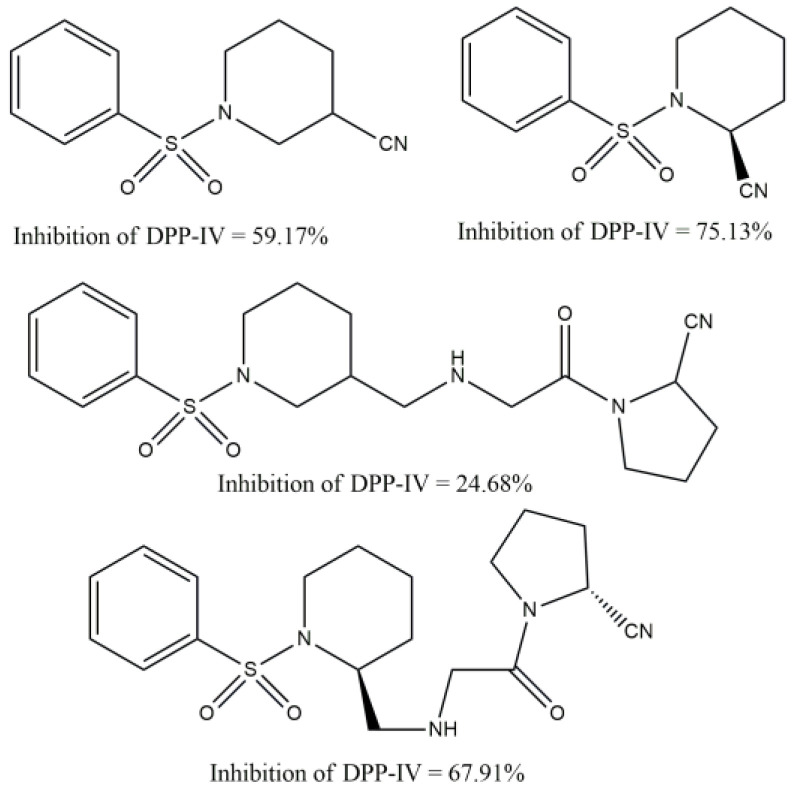
Sulfonamide derivatives of piperidine.

**Figure 28 molecules-29-03043-f028:**
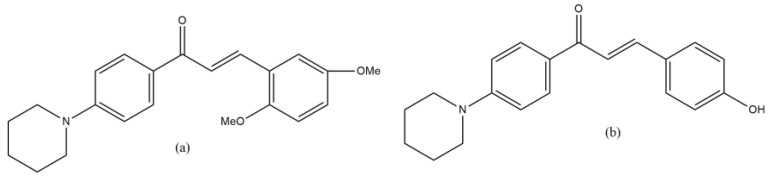
Piperidine-substituted chalcones with α-amylase inhibitory (**a**) and radical scavenging (**b**) activities.

**Figure 29 molecules-29-03043-f029:**
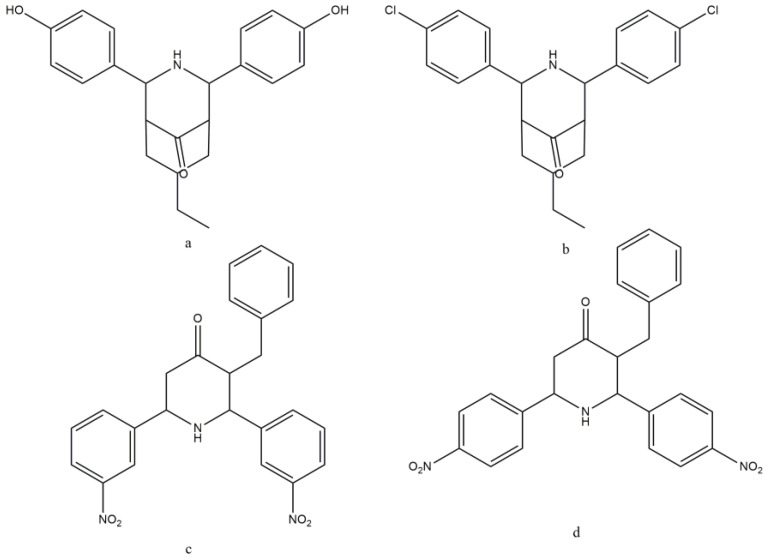
Chalcone derivatives of piperidine. (**a**) 7-ethyl-2,4-bis(4-hydroxy-phenyl)-3-azabicyclo[3.3.1]nonan-9-one; (**b**) 7-ethyl-2,4-bis(4-chloro-phenyl)-3-azabicyclo[3.3.1]nonan-9-one; (**c**) 3-benzyl-2,6-bis(3-nitrophenyl)piperidin-4-one; (**d**) 3-benzyl-2,6-bis(4-nitrophenyl)piperidin-4-one.

**Figure 30 molecules-29-03043-f030:**
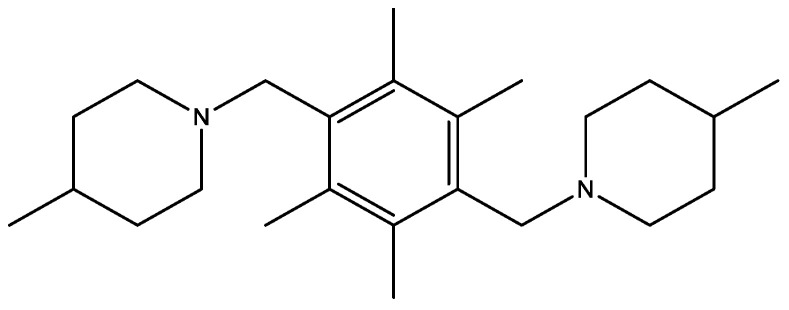
4-Methyl-1-({2,3,5,6-tetramethyl-4-[(4-methylpiperidinyl)methyl]phenyl}methyl)piperidine as α-amylase inhibitor.

**Figure 31 molecules-29-03043-f031:**
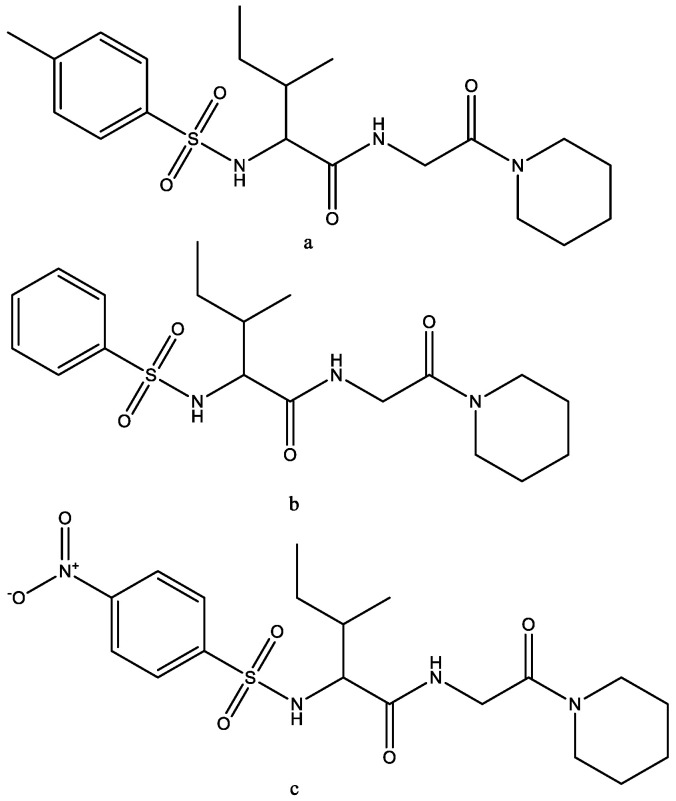
Piperidine-derived sulfonamides. (**a**) 3-Methyl-2-(4-methylphenylsulfonamido)-N-(2-oxo-2(piperidin-1-yl) ethyl) pentanamide; (**b**) 3-methyl-N-(2-oxo-2-(piperidin-1-yl)ethyl))-2-(phenylsulfonamido)pentanamide; (**c**) 3-methyl-2-(4- nitrophenylsulfonamido)-N-(2-oxo-2-(piperidin-1-yl)ethyl)pentanamide.

**Figure 32 molecules-29-03043-f032:**
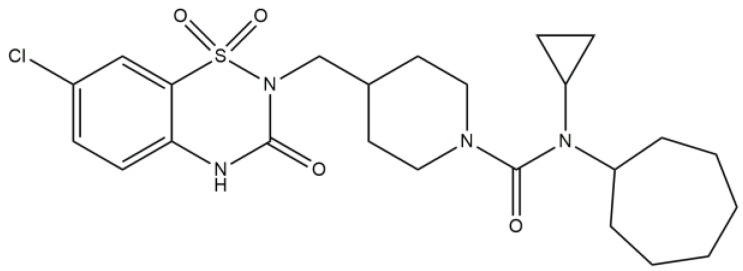
Piperidine amide and urea derivative.

## Data Availability

The data presented in this study are available upon request from the corresponding author.

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
