# Peer review of "Morpholine, Piperazine, and Piperidine Derivatives as Antidiabetic Agents"

_molecules, 2024, doi:10.3390/molecules29133043_

Round 1
Reviewer 1 Report
Comments and Suggestions for Authors
The paper titled “Morpholine, piperazine and piperidine derivatives as antidiabetic agents” shows the antidiabetic potential of the above heterocycles. I can say in advance that the work should be of interest to readers, since the fragments are quite popular and the review should receive good citations. However, I have two comments about the text:
These fragments are not completely pharmacophoric and, unlike indole, their presence does not provide a high probability of activity of the molecule. In addition, in the presented structures, the presented heterocycles are often a small component, while the pharmacoform may be the rest of the skeleton. Therefore, it seems to me important to give examples of similar compounds without these heterocycles in order to prove the main line of the story.
Second: Similar to the first, the listing within the chapters does not have any strict pattern and in-depth analysis. Appears more like a simple enumeration. Should there be any conclusions in the work - what is necessary for the manifestation of antidiabetic activity? For example, you can compare figures 6 and 18, although 6 also contains piperazines.
A small note: in the list of references,DOI sometimes in the form of a link, sometimes not.
Author Response
Answer to reviewers:
First, the authors thank the reviewers for evaluating our article and for fair comments on our work, which we tried to correct. We also ask you to make changes to the affiliation to one of the authors Tulegen Seilkhanov. It is necessary to add the word “University” (Laboratory of Engineering Profile NMR Spectroscopy, Sh. Ualikhanov Kokshetau University). Also, we ask to delete grant number AP19578051 from Funding section and to keep only one AP19676539.
Comments and Suggestions for Authors
The paper titled “Morpholine, piperazine and piperidine derivatives as antidiabetic agents” shows the antidiabetic potential of the above heterocycles. I can say in advance that the work should be of interest to readers, since the fragments are quite popular and the review should receive good citations. However, I have two comments about the text:
These fragments are not completely pharmacophoric and, unlike indole, their presence does not provide a high probability of activity of the molecule. In addition, in the presented structures, the presented heterocycles are often a small component, while the pharmacoform may be the rest of the skeleton. Therefore, it seems to me important to give examples of similar compounds without these heterocycles in order to prove the main line of the story.
Answer: We added paragraphs which prove that these heterocycles strengthen the antidiabetic properties of molecule. Please see the lines 189-199, 302-308, 339-345, 427-433, 456-457.
Second: Similar to the first, the listing within the chapters does not have any strict pattern and in-depth analysis. Appears more like a simple enumeration. Should there be any conclusions in the work - what is necessary for the manifestation of antidiabetic activity? For example, you can compare figures 6 and 18, although 6 also contains piperazines.
Answer: After Figure 18 paragraph was added. In the conclusion section also was added the sentence that manifesting the antidiabetic activity of our compounds.
A small note: in the list of references,DOI sometimes in the form of a link, sometimes not.
Answer: DOIs have been converted to a form without links
Reviewer 2 Report
Comments and Suggestions for Authors
Referee report on the manuscript submitted to the Molecules journal and entitled: ”Morpholine, piperazine and piperidine derivatives as antidiabetic agents”. The Authors of the manuscript are Darya Zolotareva, Alexey Zazybin, Anuar Dauletbakov, Yelizaveta Belyankova, Beatriz Giner Parache, Saniya Tursynbek, Tulegen Seilkhanov, Anel Kairullinova.
This is a review article where the Authors present current knowledge of morpholine, piperazine and piperidine as potential antidiabetic agents. The Authors in their reference list cited 45 publications. In the manuscript the Authors provided a current knowledge of the diabetes and the report how we can prevent this disease. In addition, they presented natural compounds containing chemical moiety decreasing the sugar level in the blood. They described several compounds containing morpholine, piperazine and piperidine moitety investigated using in vivo, in vitro assays with promising biological activity against diabetes. The reported also on in silico studies supporting the findings. The Reviewer found the study interesting and suitable for the Molecules journal.
Some proposed corrections:
Figure 1 – the quality should be improved. In the current form it is unreadable. The x and y axes should be defined.
Line 354 founded – should be found
Line 437 the format of the figure caption should be improved.
to represents should be to represent
The list of References should be checked, unified and prepared according to the journal requirements:
12 journal abbreviation missing
21 the abbreviations of J. Mol. Struct. Is written with “dots” however in the references 29,30, 43 the “dots” are missing
38 J. Diabetes instead of J. diabetes
Comments on the Quality of English LanguageThe Reviewer found the English language understandable. The Reviewer made some remarks concerning the language in the referee report.
Author Response
Answer to reviewers:
First, the authors thank the reviewers for evaluating our article and for fair comments on our work, which we tried to correct. We also ask you to make changes to the affiliation to one of the authors Tulegen Seilkhanov. It is necessary to add the word “University” (Laboratory of Engineering Profile NMR Spectroscopy, Sh. Ualikhanov Kokshetau University). Also, we ask to delete grant number AP19578051 from Funding section and to keep only one AP19676539.
Referee report on the manuscript submitted to the Molecules journal and entitled: ”Morpholine, piperazine and piperidine derivatives as antidiabetic agents”. The Authors of the manuscript are Darya Zolotareva, Alexey Zazybin, Anuar Dauletbakov, Yelizaveta Belyankova, Beatriz Giner Parache, Saniya Tursynbek, Tulegen Seilkhanov, Anel Kairullinova.
This is a review article where the Authors present current knowledge of morpholine, piperazine and piperidine as potential antidiabetic agents. The Authors in their reference list cited 45 publications. In the manuscript the Authors provided a current knowledge of the diabetes and the report how we can prevent this disease. In addition, they presented natural compounds containing chemical moiety decreasing the sugar level in the blood. They described several compounds containing morpholine, piperazine and piperidine moitety investigated using in vivo, in vitro assays with promising biological activity against diabetes. The reported also on in silico studies supporting the findings. The Reviewer found the study interesting and suitable for the Molecules journal.
Some proposed corrections:
Figure 1 – the quality should be improved. In the current form it is unreadable. The x and y axes should be defined.
Answer: The x and y axes were named.
Line 354 founded – should be found
Answer: Changed
Line 437 the format of the figure caption should be improved.
to represents should be to represent
Answer: The caption of the figure was improved, also it was added the “Figure 30” in the text.
The list of References should be checked, unified and prepared according to the journal requirements:
12 journal abbreviation missing
Answer: added journal abbreviation
21 the abbreviations of J. Mol. Struct. Is written with “dots” however in the references 29,30, 43 the “dots” are missing
Answer: dots were deleted
38 J. Diabetes instead of J. diabetes
Answer: changed
Round 2
Reviewer 1 Report
Comments and Suggestions for Authors
The authors really did a good job and took into account all the comments.
After the audit, work improved significantly. Now I can recommend it for publication.